# Investigation into Recognition Technology of Helmet Wearing Based on HBSYOLOX-s

**Teng Gao and Xianwu Zhang \***

Key Laboratory of Signal Detection and Processing, College of Information Science and Engineering, Xinjiang University, Urumqi 830000, China
* Correspondence: zxw@xju.edu.cn

**Abstract:** This work proposes a new approach based on YOLOX model enhancement for the helmet-wearing real-time detection task, which is plagued by low detection accuracy, incorrect detection, and missing detection. First, in the backbone network, recursive gated convolution ($g^nConv$) is utilized instead of traditional convolution, hence addressing the issue of extracting many worthless features due to excessive redundancy in the process of feature extraction using conventional convolution. Replace the original FPN layer in the Neck network with the EfficientNet-BiFPN layer. Realize top–down and bottom–up bidirectional fusion of deep and shallow features to improve the flow of feature data between network layers. Lastly, the SIOU cross-entropy loss function is implemented to address the issue of missed detections in crowded environments and further increase the model's detection precision. Experiments and data comparisons indicate that the modified model's average detection accuracy is 95.5%, which is 5.4% higher than that of the original network model, and that the detection speed has been dramatically increased to suit actual production requirements.

**Keywords:** safety helmet wear detection; recursive gated convolution; feature integration; SIOU



## 1. Introduction

With increasing urbanization, the building industry has experienced substantial growth. However, numerous factors threaten individuals' life safety in the complex environment of construction sites, and research has shown that most of these threats are due to head injuries. Helmets can shield the heads of construction workers, safeguarding their lives and lowering the incidence of risk. In the actual construction site, often due to the lack of safety awareness of construction personnel ignore the helmet wearing.Therefore, the intelligent supervision of the hard helmet wearing status is very important. Due to the emphasis on worker safety, a great deal of research has been undertaken in this field by a large number of experts. Traditional helmet-wearing detection tasks depend primarily on image processing, machine learning, and sensor-sensing-based techniques [1–3].

Columns include Jie [4] detecting moving things in construction sites using the background modeling approach based on standard image processing and machine learning characterizes the human behavior in the region of interest using the histogram of oriented gradient (HOG). Singh [5] utilized a sensor to put a device for detection on a helmet, enabling the physical detection of helmet use. Feng [6] uses the foreground information extracted from the collected video pictures. The foreground targets were binarized to isolate the moving objects, and the pedestrians were tagged and tracked using scale filtering to separate the travelers from other moving things. Finally, compare the chromaticity distribution of the target with the set helmet color to identify the helmet's color and realize the staff's helmet-wearing detection. All the techniques mentioned above rely on traditional approaches, albeit to varying degrees, to accomplish the effect of helmet detection.

In recent years, significant advances have been made in deep learning and intelligent industrial security. In industrial intelligence, deep-learning-based target detection technology is frequently employed. Depending on the existence or absence of a candidate region

generation process, the current deep-learning-based target identification algorithms are classified into two-stage and one-stage detection. The stage of target detection is represented by the R-CNN series based on the generation of a candidate frame, the models, such as fast R-CNN [7], faster R-CNN [8], and mask R-CNN [9]. The model based on the candidate region for quadratic correction regression yields highly accurate detection results. Unfortunately, the detection timeliness is middling, making it unsuitable for detection applications with strict timeliness requirements. One-stage target detection models are separated into YOLO [10–14] and SSD [15] series. Without creating separate candidate frames, our single-stage detection algorithms identify and position-correct these frames directly. The detection speed is increased to satisfy real-time performance requirements, but it does not filter the candidate frames, resulting in a model with very low accuracy. Consequently, we are now optimizing and enhancing the one-stage detection algorithm for helmet detection to achieve the optimal balance between accuracy and speed for helmet detection.

In this paper, we examine the challenges of erroneous detection caused by a hand-held helmet task, a long distance and a tiny target, and a low detection rate caused by dense and severe occlusion, by enhancing the YOLOX-s method and verifying it using a custom-built helmet dataset [16].

We have mainly made the following improvements on YOLOX-s: (1) To address the issue of challenging feature extraction in the target and increase detection accuracy, the HorNet module is utilized to replace the Res8 module of the backbone network. (2) The original FPN structure is replaced with BiFPN to improve feature fusion while minimizing feature loss, resolving the issue of low detection accuracy under dense occlusion. (3) The original IOU loss function is changed with the SIOU loss function to address missed detection due to extreme occlusion and further increase the model's detection accuracy.

This paper's structure is as follows: In the second section of the article, the present state of deep learning research on helmet detection is reviewed. In the third section, the updated YOLOX-s algorithm is described. The fourth portion provides the dataset and a comparison of the experimental outcomes. We conclude with an overview and summary of the entire material.

## 2. Related Work

Because the first-stage target detection algorithm has the advantages of high detection accuracy and speed, many researchers have conducted a lot of research and optimization based on the first-stage target detection model for helmet-wearing detection tasks. Wang et al. [17] used the cross-stage partial network (CSPNet) to improve the backbone network of Darknet53 based on the YOLOv3 model with a spatial pyramid set (SPP) structure. They also enhanced the multiscale prediction network with top–down and bottom–up feature fusion strategies to achieve feature enhancement. The improved model detection accuracy and speed increased by 28% and 6 fps/s, respectively, compared with the original YOLOv3. Huang et al. [18] used the improved YOLOv3 algorithm to output the predicted anchor box of the target object for pixel feature statistics and output the confidence level, followed by increasing the fourth feature map scale, enhancing the loss function, and combining the image processing pixel feature to detect whether or not the helmet is worn accurately. The detection speed and precision of the enhanced detection algorithm for the helmet detection task have been enhanced in comparison with the prior method. Zeng et al. [19] improved the model's accuracy and speed by swapping out the tedious overlap of many convolution modules in the original YOLOv4 feature pyramid for cross-stage hierarchical modules. Changing the linear transformation of the YOLOv4 feature layer output and anchor improves detection performance for helmet identification of tiny targets. Sun et al. [20] updated helmet wearing recognition method based on YOLOv5 to collect more efficient information. MCA attention mechanism is introduced to the feature extraction network, and the improved algorithm is pruned to compress the model. Wang et al. [21] added a convolutional attention module to the feature fusion network in the YOLOv5 backbone network by employing a deformable convolutional net

instead of the conventional convolution and replacing the original network's generalized cross-joint loss function with distance cross-joint loss. Lv et al. [22] proposed a method for helmet detection to enhance the YOLOX-s network. The number of training sets is increased by merging the mosaic approach with online complex sample mining (OHEM); a branch attention module is introduced to the prediction side of the model (prediction), and a novel cosine annealing algorithm is presented to minimize the model's convergence time. The improved helmet detection method improves the mAP, accuracy, and recall rate of helmet detection compared with those before the improvement.

Most of the scholars above mainly optimize the detection speed and accuracy of the model, but they do not address the actual problems faced in the task of helmet-wearing detection, so this study addresses the four issues of hand-held helmets prone to wrong inspection, the long-range small target phenomenon, the severe obstruction problem, and the crowding phenomenon based on YOLOX-s. The problem is solved and optimized.

## 3. Methodology

### 3.1. Improve Backbone

Convolutional neural networks can improve the accuracy of a model by incorporating an attention mechanism that allows the model to concentrate on features that require valuable semantic information because the Swin transformer [23], with its multiheaded attention mechanism, has been incredibly successful in the field of vision. Even so, the addition of the attention mechanism will have the problems of increasing the complexity of the model, decreasing the detection speed, and expanding the model parameters.

To address this issue, R et al. [24] proposed a HorNet module in 2022, summarizing the significant benefits of self-attention behind transformers. The diagram depicts the structure in Figure 1, in which the HorNet structure is divided into the main framework of the model and the recursive gated convolution part. The input features first enter the layer norm layer for the first time for normalization to speed up the feature processing. Next, the features are extracted using the recursive gated convolution part. The recursive gated convolution part is shown on the right of the figure below. The elements of interest to the object of study are represented in red, and the surrounding area is light gray. The design using gated convolution and recursion effectively achieves an arbitrary order of spatial interaction between the standard focal features and their surrounding regions. After extracting the features more into the layer norm layer for normalization, the final classification is performed using multilayer perceptron (MLP) to complete the enhanced feature extraction process.

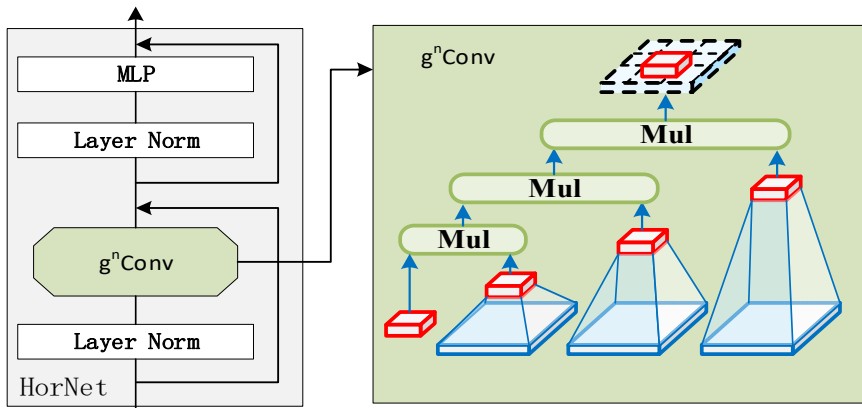

**Figure 1.** Network architecture of HorNet.

In contrast to the approach of adding attention mechanisms to the model, the implementation of $g^n$Conv of the HorNet module avoids the secondary complexity of self-attentiveness, increases the channel width during the execution of spatial interactions, and is capable of achieving high-order spatial interactions with minimal complexity. Second,

the two-order exchanges in self-attention are extended to an arbitrary order, enhancing the model's predictive capacity. The recursive gated convolution inherits the translation invariance of the conventional convolution, therefore avoiding the asymmetry produced by local attention and introducing a beneficial inductive bias for the target detection task.

HorNet is used to replace Res8 in the backbone network, which decreases the network model's complexity, improves the networks' adaptive capabilities through weight assignment and information filtering, and increases the model's computing performance. More valuable network training information is retrieved from a vast amount of feature information, which is utilized to improve the problem of challenging feature extraction in long distances, tiny targets, and dense states to increase model accuracy.

### 3.2. Feature Enhancement Network

As the features are continuously fused, the image's semantic information is strengthened, while the number of detailed features decreases. Many recent studies employ multiscale feature fusion of images, including elements at multiple levels, to improve the model's detection accuracy. In 2019, the Google team proposed the BiFPN top–down and down–top bidirectional weighted feature pyramid network [25]. The network structure is shown in Figure 2.

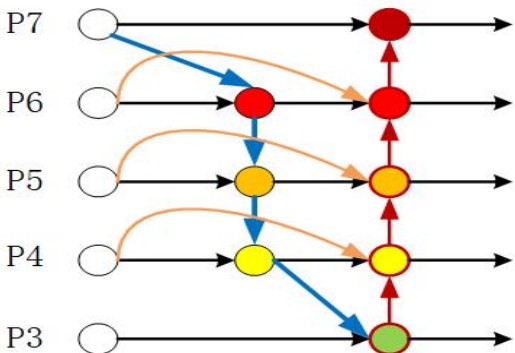

**Figure 2.** BiFPN structural diagram.

In the above structure diagram, the blue portion is the top–down pathway, which conveys the semantic information of the high-level features; the red part is the bottom–up pathway, which represents the location information of the low-level features; and the orange portion is a new edge added between the input node and the output node in the same layer as mentioned in the second point above.

The original FPN feature fusion module is replaced with a BiFPN network structure with bidirectional weighting to collect more details and exclusive domain features at multiple levels and enhance the network's ability to extract features at different sizes. Maintaining the semantic information transfer based on the improved transfer of location detail information is advantageous for the feature map to improve the helmet detection performance of the YOLOX-s model. Figure 3 depicts the overall structure of YOLOX-s after enhancing the HorNet module and BiFPN structure. The primary model structures that have been improved are shown in red boxes.

### 3.3. Improve Loss Function

To further improve the accuracy of the helmet-wearing detection task, this paper uses the SIOU loss function to replace the bounding box loss function of the predicted Reg. In 2022, Gevorgyan et al. [26] proposed the SIOU loss function algorithm to consider further the vector angle between the actual frame and the predicted frame and redefine the associated loss function. The SIOU loss function contains four components, angle cost, distance cost, shape cost, and iou cost. Compared with the original GIOU loss function,

SIOU is more consistent with the regression mechanism of the prediction box and makes the generation of bounding boxes more stable. The following formula defines SIOU:

$$LOSS_{SIOU} = 1 - IOU + \frac{\Delta + \Omega}{2} \tag{1}$$

$$\Omega = \sum_{T=W.H} \left(1 - e^{-Wt}\right)^{\theta} = \left(1 - e^{-Ww}\right)^{\theta} + \left(1 - e^{-Wh}\right)^{\theta} \tag{2}$$

$$Ww = \frac{\left|\mathrm{w} - w^{gt}\right|}{\max(w - w^{gt})}, Wh = \frac{\left|\mathrm{h} - h^{gt}\right|}{\max(h - h^{gt})} \tag{3}$$

$$\Delta = \sum_{t=x.y} \left(1 - e^{-\gamma \ell_t}\right) = 2 - e^{-\gamma \ell_x} - e^{-\gamma \ell_y} \tag{4}$$

$$\ell_x = \left(\frac{b_{cx}^{gt} - b_{cx}}{cw}\right), \ell_y = \left(\frac{b_{cy}^{gt} - b_{cy}}{ch}\right) \tag{5}$$

$$\Lambda = \cos\left(2 \times \left(\arcsin\left(\frac{c_h}{\sigma}\right) - \frac{\pi}{4}\right)\right) \tag{6}$$

$$\sigma = \sqrt{\left(b_{cx}^{gt} - b_{cx}\right)^2 + \left(b_{cy}^{gt} - b_{cy}\right)^2} \tag{7}$$

$$c_h = \max\left(b_{cy}^{gt}, b_{cy}\right) - \min\left(b_{cx}^{gt}, b_{cx}\right) \tag{8}$$

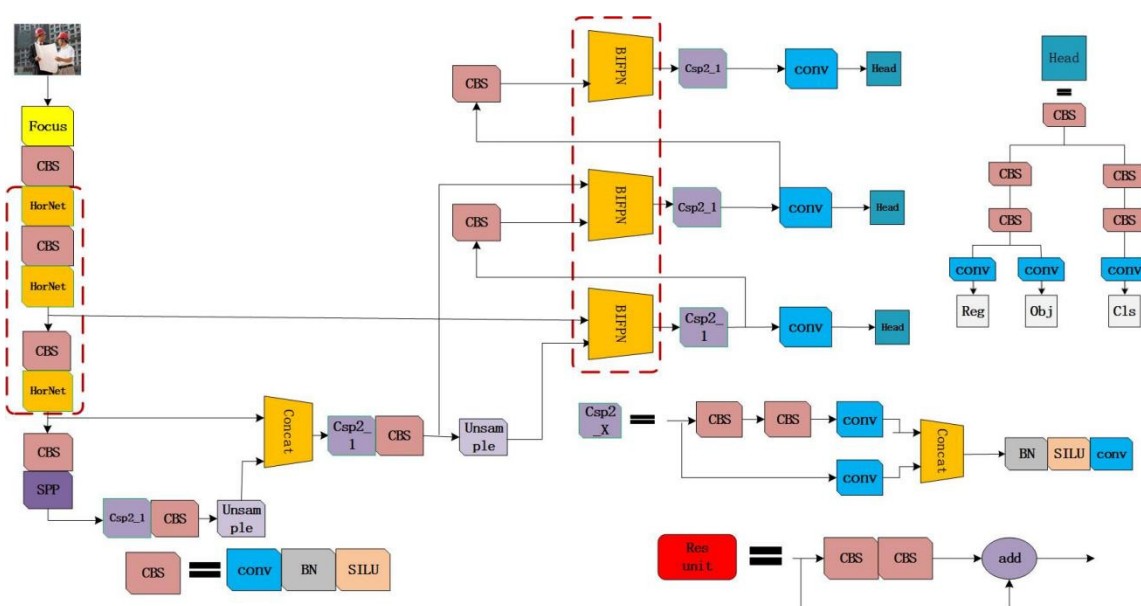

**Figure 3.** Structure of improved YOLOX-s.

In the preceding equation, IOU represents the cross-entropy loss function, whereas $\Delta$ and $\Omega$ represent the distance and shape costs, respectively. In Equation (2), $w, h, \mathrm{w}^{gt}, \mathrm{h}^{gt}$ are the predicted and real box widths and heights, which regulate the degree of attention to the shape, which is normally between 2 and 6. In Equation (4), $\gamma = 2 - \Lambda$, $c_w, c_h$ represent the minimum outer rectangle width and height of the real frame and the predicted frame, respectively, where $\sigma$ is the distance between the center points of the real frame and the predicted frame, which is calculated in the code using the Pythagorean theorem. $b_{cx}^{gt}, b_{cy}^{gt}$ are the coordinates of the real frame's center. $b_{cx}, b_{cy}$ are the center of the prediction frame's coordinates.

## 4. Experiments and Analysis

In this paper, to train and validate our improved model, the hardware environment of the experimental platform used is GPU: NVIDIA GeForce GTX 3090; CPU: Intel® Core™ i7-10750H; memory: 128 GB; and video memory: 24 GB.

### 4.1. Datasets

There currently needs to be a standardized dataset for helmet detection jobs. To include as many usage scenarios as possible and improve the generalization ability of the model, 6713 images were obtained through targeted screening of the open-source dataset SHWD, web image crawling, and construction site monitoring screenshots, and then manually labeled into Pascal VOC format using the labeling tool. The handmade helmet-wearing detection dataset has 41,198 tags categorized as helmet, wearing helmet, and not wearing helmet. A ratio of 8:2 was used to divide the training set and test set into training set and test set, with 5370 training set images and 1343 test set pictures acquired.

### 4.2. Evaluation Criteria

This research uses precision (P), recall (R), AP, and mean average precision (mAP) as model performance evaluation indices to assess the model effect. The following equations determine P, R, AP, and mAP:

$$P = \frac{TP}{TP + FP} \tag{9}$$

$$R = \frac{TP}{TP + FN} \tag{10}$$

$$AP = \int_0^1 P(t)dt \tag{11}$$

$$F1 = \frac{2P * R}{P + R} \tag{12}$$

$$mAP = \frac{\sum\limits_{n=0}^{N} AP_n}{N} \tag{13}$$

In the above equation, TP stands for actual positive sample, and PPS stands for anticipated positive example, commonly known as true positive. FN means genuine positive example, also known as expected negative selection (false negative). FP means simple negative sample, also known as anticipated positive sample (false positive). TN means true negative sample, also known as anticipated negative sample (true negative). The last statistic for model evaluation is F1, which is the summed average of precision and recall. The *mAP* is the mean of the average accuracy of all detection categories and is used to assess the detection model's overall performance.

### 4.3. Ablation Experimental Analysis

This work is trained without using the initialized weights to test the performance of the upgraded YOLOX-s model and better understand the detection effect of each improved approach. To ensure the effectiveness of the enhanced method in the experiments, epochs are set to 100. The batch size is set to 32. The input image size is 640 × 640. The improved scheme is depicted in Table 1, and the P, R, F1, mAP, and FPS indicators derived from a comparison of experiments conducted with various improved approaches are provided in Table 2. Figure 4 depicts a more visual representation of the training effect of each improvement approach on mAP boosting.

The structural enhancement techniques for YOLOX-s are detailed in Table 1. In Table 1's HorNet method, the HorNet module substitutes the Res8 module in the original Darknet53 to improve the network's feature extraction capabilities, detection accuracy, and speed. In the HorNet+BiFPN scheme, the FPN structure of Neck is replaced with top–down and bottom–up bidirectional weighted feature fusion network structures, based on the

improved HorNet method, to address the issue of partial speech information loss during feature fusion and enhance model robustness. In the HorNet+BiFPN+SIOU scheme, the original loss function is replaced with the SIOU loss function to increase the regression accuracy of the target and the detection accuracy in the situation of dense occlusion to improve the model detection accuracy further.

**Table 1.** Various module combinations are displayed; "√" indicates the selected module.

| Model | HorNet | BiFPN | SIOU |
|---|---|---|---|
| YOLOX-S | | | |
| HorNet | √ | | |
| HorNet + BiFPN | √ | √ | |
| HorNet + BiFPN + SIOU | √ | √ | √ |

**Table 2.** Results of several improved ablation methods.

| Model | P (%) | R (%) | F1 (%) | mAP | FPS |
|---|---|---|---|---|---|
| YOLOX-S | 90.8 | 91.2 | 91.0 | 90.1 | 37.5 |
| HorNet | 91.4 | 92.3 | 91.9 | 91.6 | 36.4 |
| HorNet + BiFPN | 94.5 | 91.1 | 92.8 | 94.8 | 39.3 |
| HorNet + BiFPN + SIOU | 95.6 | 92.4 | 94.1 | 95.5 | 41.3 |

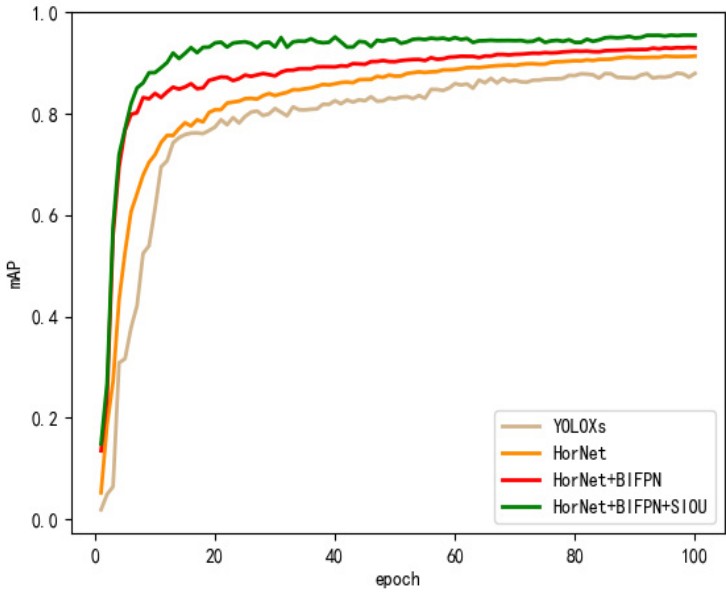

**Figure 4.** mAP curves of different improvement methods.

According to the improved results demonstrated in Table 2, we analyze the ablation experiment with an input image size of 640 × 640, and we can see that the results of P, R, F1, and mAP are relatively low in the original YOLOX-s algorithm. After using the HorNet module instead of the original Res module, using the self-attention mechanism to make the network focus on the information that needs attention, the values of P, F1, and mAP are improved by 0.6%, 0.8%, and 1.5%, respectively. Compared with the unimproved model, the HorNet module can effectively improve the detection accuracy of the model. The HorNet+BiFPN scheme improves the values of P, F1, and mAP by 3.1%, 0.9%, and 3.2%, respectively, which shows a significant improvement of the network effect after the improved two-way weighted feature fusion by experimental data. In the HorNet+BiFPN+SIOU scheme, P, F1, and mAP are increased by 1.1%, 1.3%, and 0.7%, respectively, while keeping the detection speed unchanged. The SIOU loss function is introduced to solve the problem of low detection accuracy caused by severe occlusion, and the effectiveness of this method

is verified. This program also adopts the final scheme, and the HorNet+BiFPN+SIOU scheme is renamed HBSYOLOX-s.

Figure 4 depicts the mAP training process, with the horizontal axis representing the training batches and the vertical axis showing the mAP change results to compare each approach's improvement effect visually. Figure 4 depicts the success of each improvement approach, where the green curve shows the HBSYOLOX-s scheme, and the improvement is attributed to the other methods.

### 4.4. Comparison with Other Models

To verify the reliability of the improved YOLOX algorithm, we primarily compared the mainstream target detection algorithms at one stage with the algorithm proposed in this paper, including fast SSD, YOLOv3, YOLOv4, YOLOv5-s, YOLOX-s, and the HBSYOLOX-s algorithm proposed in this paper; the results of these comparisons are shown in Table 3.

**Table 3.** Performance comparison under different models.

| Model | P (%) | R (%) | F1 (%) | mAP | FPS |
|---|---|---|---|---|---|
| SSD | 82.6 | 81.3 | 81.9 | 82.6 | 25.8 |
| YOLOv3 | 83.4 | 82.1 | 82.7 | 84.2 | 31.6 |
| YOLOv4 | 87.5 | 86.4 | 86.9 | 87.6 | 33.4 |
| YOLOv5-s | 89.6 | 84.6 | 87.0 | 89.3 | 32.2 |
| YOLOX-s | 90.8 | 91.2 | 91.2 | 90.1 | 37.5 |
| OURS | 95.6 | 92.4 | 94.1 | 95.5 | 41.3 |

Comparing the precision, recall, F1, mAP, and FPS values of each approach, as shown in Table 3, demonstrates that the improved HBSYOLOX-s algorithm described in this study is primarily due to SSD, in addition to other models from the YOLO family. YOLOX's precision and recall are more significant than YOLOv5's due to its innovative use of decoupling heads. As a result, the technique outlined in this paper improves the YOLOX model, which has achieved superior results by substituting modules such as HorNet, BiFPN, and SIOU, substantially enhancing the model's performance indices. The model's precision, durability, and detection speed have improved for wider practical use.

### 4.5. Detection Results in Application Scenarios

To evaluate the identification accuracy and resilience of the revised method in complicated building sites, we demonstrate the detection effect in a subset of complex natural situations. As seen in Figure 5, the detection effects are shown for four states: easily misidentified, tiny target, severe occlusion, and thick picture. Because there are several portable helmets on the actual construction site shown in Figure 5a, the general model incorrectly determines whether or not workers are wearing helmets. According to Figure 5a, the better model is distinguishable, and the helmet's location and the person's state are shown in red and blue boxes, respectively. In Figure 5b, the vast distance in the picture makes it difficult to identify tiny things, and it is possible to experience the issue of missed detection. In this study, we apply the HorNet module to improve feature extraction, enabling the detection of tiny targets at great distances. Figure 5c,d demonstrates that the issue of dense people and severe obstruction may be resolved using an algorithm with an enhanced SIOU loss function. Based on the results of various scenario tests, the improved algorithm has an excellent detection impact.

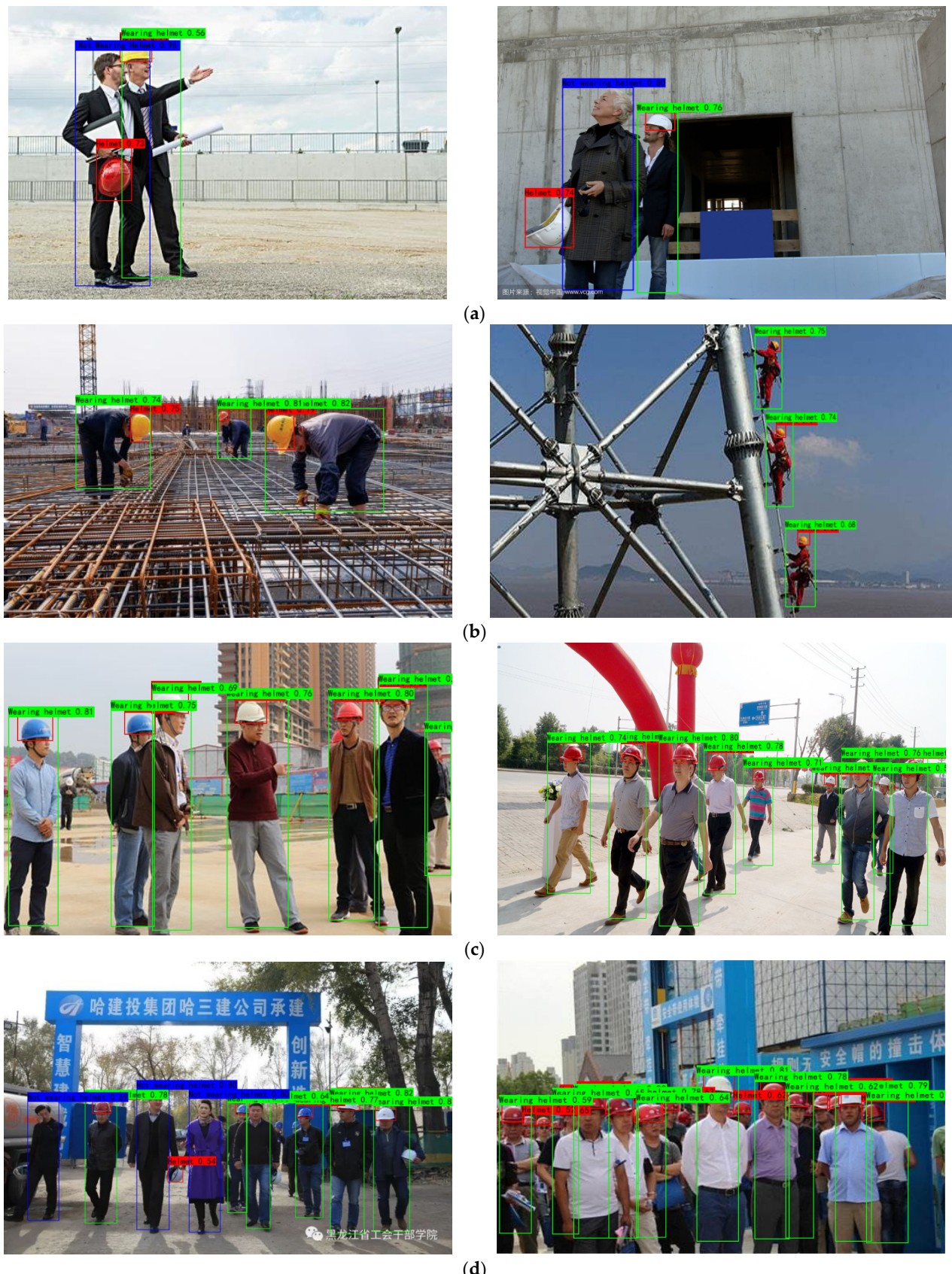

**Figure 5.** (**a**) Hand-held helmet prone to wrong inspection, (**b**) long-range small target phenomenon, (**c**) severe obstruction problem, (**d**) crowding phenomenon.

## 5. Conclusions

Currently, there are public datasets, such as COCO and VOC(07+12), in the target detection field, which are only annotated with the names of people and other animals to improve the detection accuracy of the generic model. However, there is no collection and labeling for context-specific detection tasks, such as helmet wear detection, so the current public datasets need to be more flexible to meet detection requirements. In this study, we build an HBSYOLOX-s model for helmet-wearing detection that can accurately recognize the helmet-wearing circumstance and improve the flexibility and robustness of the model, as shown below.

1. This study replaces the Res8 module of DrakNet with the HorNet module using recursive gated convolution to build a more comprehensive feature map. This is performed to extract all essential features from the target. Meanwhile, eliminating the attention mechanism eliminates superfluous model parameters and provides a good foundation for feature fusion.
2. To fully fuse the extracted important features and prevent the loss of semantic information during the fusion process, the original FPN network is replaced with a BiFPN two-way weighted feature fusion network, which assigns higher feature weights to the large amount of helpful feature information obtained in the previous step. It eliminates the issues of false detection in extreme occlusion and the low detection rate of small targets at great distances.
3. On top of the improved model, the original loss function is replaced with SIOU, which further enhances the model's detection accuracy, and the three improved approaches are then integrated to accomplish the task of helmet-wearing detection.

Ablation experiments demonstrate that the enhanced approach has an mAP of up to 95.5%, which is 5.4% higher than before the enhancement, and a detection speed of up to 41.3 fps/s. Compared with the YOLOX-s and YOLOv5-s models, the network model parameters are more affluent, the network model's flexibility and resilience are improved, and the model suggested in this research has higher average detection accuracy and speed. In conclusion, the HBSYOLOX-s approach is viable and effective for the detection of helmet use. The improved algorithm was tested under extreme weather conditions, such as high temperatures, and all had good detection results, complete as required for practical detection tasks.

In the future, with the continued development of artificial intelligence algorithms, we will be able to continue studying more outstanding models from the perspective of one-stage target identification algorithms and applying them to helmet-wearing detection jobs.

**Author Contributions:** Conceptualization, T.G.; data curation, T.G.; investigation, X.Z. and X.Z.; methodology, T.G.; software, T.G.; validation, X.Z.; visualization, T.G.; writing—original draft, T.G.; writing—review and editing, T.G. All authors have read and agreed to the published version of the manuscript.

**Funding:** This work was supported in part by the Natural Science Foundation of the Xinjiang Uygur Autonomous Region under Grant 2022D01C61 and in part by the "Tianchi Doctoral Program" Project of the Xinjiang Uygur Autonomous Region under Grant TCBS202046.

**Institutional Review Board Statement:** Not applicable.

**Informed Consent Statement:** Not applicable.

**Data Availability Statement:** The data presented in this study are available on request from the corresponding author.

**Conflicts of Interest:** The authors declare no conflict of interest.

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
