# Peer review of "Investigation into Recognition Technology of Helmet Wearing Based on HBSYOLOX-s"

_applsci, doi:10.3390/app122412997_

Round 1

Reviewer 1 Report

I reviewed the manuscript tittle Investigation Into Recognition Technology of Helmet Wearing 2

Based on HBSYOLOX-s

The paper is interesting and discusses an import topic for many parties and beneficiaries. High quality paper with rich content

Many readers practitioners can benefit from this paper. However, the authors need to consider the following changes and comments to make the paper ready for publication some

Introduction

Authors need to check the citation and make sure all references have been cited correctly

Related work

Because the first-stage target detection method maintains detection accuracy while 83 vastly enhancing detection speed, other researchers have produced numerous enhance- 84 ments based on this approach

Authors must introduce this first

Please start with a general statement then move narrow the focus to the point you mentioned

In the first statement. Literature review usually starts from the main point and concern and flaw gradually

Language and structure

Please rewrite this part and fix the language

Most of the above researchers are studying improving the detection accuracy and 114 speed, but there needs to be more research for the case of motion

 with

Methodology

Figure 1. network architecture of HorNet 141

This must be followed by clear explanation of its rationale a

Figure 3. Structure of improved YOLOX-s 206

This figure is big and complex, authors  might need to simplify that

Conclusion

Please cite some examples for this statement

Most solutions offered for public datasets in deep learning target detection algo- 400 rithms are not flexible to particular detection requirements

Its is recommended that authors provide some implications for their results and conductions. Such implications will help practitioners utilize their research and understand better how this research can be used in different situations 

One slight concern about the final results, whether such product and conclusions will operate well in different locations and conditions such as temperature, wind, hot and cold areas

Author Response

Dear experts and editors,

First of all, thank you very much for your knowledge of my thesis work, your opinions and comments have been very useful in guiding me to revise my thesis according to your valuable comments, and the specific revisions are as follows.

where all changes are marked in the original

In the related work section

The citation has been checked and corrected

Corrected to:

Because the first-stage target detection algorithm has the advantages of high detection accuracy and speed, many researchers have done a lot of research and optimization based on the first-stage target detection model for helmet-wearing detection tasks.

The  language fix is :

Most of the scholars above mainly optimize the detection speed and accuracy of the model, but they do not address the actual problems faced in the task of helmet-wearing detection, so this study addresses the four issues of hand-held helmets prone to the wrong inspection, Long-range small target phenomenon, Severe Obstruction Problem, Crowding phenomenon based on YOLOX-s The problem is solved and optimized

Amend as follows:

The diagram depicts the structure in Figure 1, in which the HorNet structure is divided into the main framework of the model and the recursive gated convolution part. The input features first enter the layer norm layer for the first time for normalization to speed up the feature processing. Next, the features are extracted using the recursive gated convolution part. The recursive gated convolution part is shown on the right of the figure below. The elements of interest to the object of study are represented in red, and the surrounding area is light grey. The design using gated convolution and recursion effectively achieves an arbitrary order of spatial interaction between the standard focal features and their surrounding regions. After extracting the features more into the layer norm layer for normalization, the final classification is performed using Multilayer Perceptron (MLP) to complete the enhanced feature extraction process.

Simplification of Figure 3

We have simplified the structure of YOLOX by removing the general knowledge structure of Focus and SPP and making the structure of YOLOX clearer and more organized for researchers and others who need to use and improve it.

Amend as follows:

Currently, there are public datasets such as COCO and VOC(07+12 )in the target detection field, which are only annotated with the names of people and other animals to improve the detection accuracy of the generic model. However, there is no collection and labeling for context-specific detection tasks such as helmet wear detection, so the current public datasets need to be more flexible to meet detection requirements.

The improved algorithm was tested under extreme weather conditions, such as high temperatures, and all had good detection results, complete as required for practical detection tasks.

The detailed improvements are given in the PDF file

Reviewer 2 Report

At first, I sincerely apologize for my delayed report.

This paper proposes the new scheme for the real-time detections of persons who wear helmet with good characteristics of high-accuracy and high-speed, which is called HBSYOLOX-s. The proposed scheme is originated from YOLOX-s, but HorNet module, BIFPN (bi-directional feature pyramid network) structure, and SIOU (sensitive intersection over unit) loss function are adopted to optimize the detection efficiency for the above purpose. YOLO (you only look once) algorithm is a convolutional neural network (CNN) for object detections. The proposed algorithm can obtain higher detection accuracies than the ones of other algorithms. The results in this paper may appear sounds for the readers of Applied Sciences. However, the reasons why the authors chose the above schemes are unclear. Thus, the reviewer would like to recommend “Accept after minor revision.” The points for the required revisions are summarized in the field of “Comments and Suggestions for Authors.”

Author Response

Dear experts and editors:

First of all, thank you very much for your expert knowledge of my thesis work. Your comments and reviews have been very useful in guiding me to revise my thesis in accordance with your valuable advice, as follows.

At the first:

The YOLOX model was chosen as the baseline model because it is currently one of the most advanced algorithms in the field of target detection, because of the innovative use of structures such as anchorless frames and decoupling heads, which have achieved good detection results in the field of target detection, and therefore this study improves on the YOLOX model.

Second reason:

The HorNet module is first used to replace the Res8 module, as in contrast to the approach of adding attention mechanisms to the model, the implementation of the gnConv of HorNet module avoids the secondary complexity of self-attentiveness, increases the channel width during the execution of spatial interactions, and is capable of achieving high-order spatial interactions with minimal complexity. Second, the two-order exchanges in self-attention are extended to arbitrary order, enhancing the model's predictive capacity. The recursive gated convolution inherits the translation invariance of the conventional convolution, so avoiding the asymmetry produced by local attention and introducing a beneficial inductive bias for the target detection task.

Three reason:

BIFPN features efficient bi-directional cross-scale connectivity and weighted feature map fusion

The original FPN feature fusion module is replaced with a BIFPN network structure with bi-directional weighting to collect more details and exclusive domain features at multiple levels and enhance the network's ability to extract features at different sizes. Maintaining the semantic information transfer based on the improved transfer of location detail information is advantageous for the feature map to improve the helmet detection performance of the YOLOX-s model.

Fourth reason:

The original loss function did not take into account the mismatch between the desired true frame and the predicted frame during feature extraction, which resulted in slow and inefficient convergence. The new loss function SIOU is therefore utilised in the helmet wear detection task, which takes into account the vector angle between the required regressions and redefines the penalty metric. The application to the detection task helps to improve the training speed and inference accuracy of the model.

Summary:

The proposed scheme is originated from YOLOX-s, and HorNet module, BIFPN (bi-directional feature pyramid network) structure, and SIOU (sensitive intersection over unit) loss function are adopted to optimize the detection efficiency for the above purpose.
